# Anti-Influenza Virus Potential of Probiotic Strain *Lactoplantibacillus plantarum* YML015 Isolated from Korean Fermented Vegetable

Rajib Majumder [1,2,3,†], Md Badrul Alam [4,†], Keshav Raj Paudel [1], Khandaker Asif Ahmed [2,5], Hari Prasad Devkota [6,7,8], Sang-Han Lee [4], Philip M. Hansbro [1,*] and Yong-Ha Park [3,*]

1 Centre for Inflammation, Centenary Institute and University of Technology Sydney, Faculty of Science, School of Life Sciences, Sydney, NSW 2007, Australia
2 Applied Biosciences, Macquarie University, Sydney, NSW 2109, Australia
3 Department of Applied Microbiology and Biotechnology, Yeungnam University, Gyeongsan 38541, Korea
4 Department of Food Science and Biotechnology, Graduate School, Kyunpook National University, Daegu 37224, Korea
5 Australian Centre for Disease Preparedness (ACDP), CSIRO, Geelong, VIC 3220, Australia
6 Graduate School of Pharmaceutical Sciences, Kumamoto University, 5-1 Oe-honmachi, Chuo-ku, Kumamoto 862-0973, Japan
7 Headquarters for Admissions and Education, Kumamoto University, Kurokami, 2-39-1, Chuo-ku, Kumamoto 860-8555, Japan
8 Pharmacy Program, Gandaki University, Pokhara 33700, Nepal
* Correspondence: philip.hansbro@uts.edu.au (P.M.H.); peter@ynu.ac.kr (Y.-H.P.); Fax: +82-53-813-4620 (Y.-H.P.)
† These authors contributed equally to this work.

**Abstract:** Lactic acid bacteria are one of the potential natural remedies used worldwide, commonly known as probiotics. Here, the aim of this research investigation was to isolate a probiotic *Lactobacilli* strain, YLM015, from the popular Korean fermented vegetable "Kimchi" and to evaluate its anti-viral potential against influenza virus A (IFVA) H1N1 using the MDCK cell line in vitro, and in embryonated eggs in ovo. The YML015 strain was selected from among the 1200 *Lactobacilli* isolates for further studies based on its potent anti-viral efficacy. YML015 was identified and characterized as *Lactoplantibacillus plantarum* YML015 based on the 16S rRNA gene sequencing and biochemically with an API 50 CHL Kit. In ovo assay experienced with embryonated eggs and the hemagglutination inhibition method, as well as cytopathogenic reduction assay, was performed individually to observe anti-influenza viral activity of YML015 against influenza virus A H1N1. Additionally, YML015 was classified for its non-resistance nature as safe for humans and animals as confirmed by the antibiotic susceptibility (MIC) test, cell viability, and hemolysis assay. The heat stability test was also experienced by using different heat-treated cell-free supernatant (CFS) samples of YML015. As a result, YML015 showed highly potent anti-viral activity against influenza virus A H1N1 in vitro in the MDCK cell line. Overall findings suggest that anti-influenza viral activity of *L. plantarum* YML015 makes it a potential candidate of choice for use as an influential probiotic in pharmacological preparations to protect humans and animals from flu and viral infection.

**Keywords:** probiotic; *L. plantarum* YML015; antiviral activity

## 1. Introduction

Nowadays, probiotics have gained major attention due to their diverse range of therapeutic potential. Probiotics are defined as live microorganisms that, when administered in adequate amounts, confer health benefits on the host (WHO/FAO, 2002). The efficacy of probiotic bacteria is greatly influenced by their functional properties, such as antimicrobial activity, persistence in the gastrointestinal tract (GIT) for intestine-targeted probiotics, and

immunomodulatory properties [1,2] In the gut, probiotics affect balancing and restoration of the gut microbiota, protection against pathogens, immunomodulation, and maintenance of intestinal barrier integrity [3]. Studies reported that probiotics are frequently used in dietary supplements, food, infant formula formulations, and medical devices [4,5]. In addition, probiotics have significant potential as therapeutic options for a variety of diseases, mainly gastrointestinal diseases (including acute infectious diarrhea, antibiotic-associated diarrhea, ulcerative colitis, irritable bowel syndrome, functional gastrointestinal disorders, or necrotizing enterocolitis), as well as extra-intestinal disorders, such as hepatic encephalopathy [6,7]. Probiotics not only show the healing effect as live compounds, but they also produce the same pharmacological effect as dead microorganisms, confirming that microbial metabolites present in probiotics have the ability to prevent various types of chronic diseases, including allergy, acute infectious diarrhea, and bowel disease inflammation [8]. *Bifidobacterium* and *Lactobacillus* microorganisms are most frequently used as probiotics. Interestingly, previous research has been reported that *lactobacillus* as a probiotic has potential effect on bacterial infectious diseases caused by *Streptococcus pyogenes* and *Streptococcus pneumoniae* [9,10]. In addition, studies found that the early-stage cold has been inhibited using *Lactobacillus* with yogurt [11]. However, not all probiotics display the same properties; hence, careful selection of specific strains based on their claimed beneficial effects is needed.

Influenza, an infectious disease of birds and mammals, is caused by RNA viruses of the family Orthomyxoviridae (commonly called the influenza viruses). Over the recent pandemic years, between three and five million people have been infected and between 250,000 and 550,000 have died due to the seasonal spread of this deadly influenza virus around the world [12–15]. Humans, birds, and pigs are the common host for influenza virus. Based on two specific proteins called hemagglutinin (HA) and neuraminidase (NA), there are different subtypes of influenza viruses [16]. Influenza virus A H1N1, which affects humans, has some common symptoms, such as fever, pharyngitis, muscle pain, severe headache, include chills, coughing, and weakness [13,14]. It also causes pneumonia, which might be fatal in more serious cases, especially for young children and elderly people [17]. So far, the management authority of influenza disease has been given optimum priority worldwide to control the spread of influenza viruses. In general, people are vaccinated against influenza as a practical and potentially anti-viral therapy. However, considering the limited efficacy of the current vaccination programs [18] and imposed restrictions on the use of antiviral drugs (those that have severe side effects) [19], the development of probiotic-based natural anti-viral therapies is in high demand. A few studies have demonstrated that probiotics of some lactic acid bacterial strains were able to protect against infectious diseases [20,21] and have the anti-allergic effects on immune diseases in mice [22,23] and humans [24,25]. Yogurt fermented with *Lactobacillus* was shown to reduce the cases of catching cold in the healthy elderly [11] and to prolong the survival periods of mice with influenza virus infection [24]. Despite this current situation, there is an alarming demand arising for the development of natural antiviral therapy of probiotic supplements against influenza virus, which might be safe and free from any adversary side effects as the lactic acid bacteria (LAB) and/or probiotics are classified as generally safe (GRAS) by US-FDA. Therefore, the primary objective of this research is to isolate, identify, and characterize the novel probiotic strain *L. plantarum* YML015 and to evaluate its potent antiviral efficacy against influenza virus A H1N1 in vitro and in ovo.

## 2. Materials and Methods

### 2.1. Isolation and Identification of L. plantarum YML015

2.1.1. Isolation and Stock Preparation Process of *L. plantarum* YML015

A total of one 1200 Lactic Acid Bacteria (LAB) strains were isolated from various home-made traditional Korean fermented vegetables called "Kimchi" using a serial dilution method of 10-fold dilutions with PBS as described previously with slight modifications [26]. 100 μL of diluted sample was spared and placed on Bromcresol purple agar (BCP agar) medium

(Merck, Darmstadt, Germany) for primary identification based on colony morphology and color changes around the single colonies. Prior to final stock preparation, the pure single colony isolated from the BCP agar plates were cultured again on de Man–Rogosa–Sharpe agar medium (MRS agar; Merck, Darmstadt, Germany). The agar plates were incubated at 37 °C inside the anaerobic incubator for 24 h (MRS agar) and 48 h (BCP agar) [27]. Cell-free supernatants (CFS) of bacterial isolates were processed by culturing in MRS broth at 37 °C for 24 h, followed by centrifugation (10,000× $g$ for 10 min) and filtration (0.22 μm filter; Sartorius Stedim Biotech, Goettingen, Germany). Furthermore, bacterial stock was prepared by 30% glycerol and stored as a working cell bank at −80 °C for further use [28,29].

### 2.1.2. DNA Extraction, PCR, and 16S rRNA Gene Sequencing of *L. plantarum* YML015

As it flourished on the preliminary antiviral screening against IFVA H1N1, *L. plantarum* YML015 was selected as the best LAB candidate and further subjected to 16S rRNA gene sequencing to confirm the identification. Before the sample was sent for Sanger sequencing, DNA extraction was performed using a pure single colony of *L. plantarum* YML015 cultured on MRS agar (Sigma-Aldrich, St. Louis, MO, USA) medium after 24 h of incubation at 37 °C. Whole DNA extraction was carried out using the Isolate II genomic DNA kit from Bioline, USA following the protocol provided by the manufacturer, and extracted DNA were quantified by Nano-drop (Eppendorf, Hamburg, Germany). Standard combination of universal Primers (27f/1492r primers) were used to continue initial PCR amplifications [30]. All PCR amplifications in the present study were performed in an Eppendorf thermocycler (Eppendorf, Germany). The PCR profile included an initial denaturing step at 95 °C for 2 min, followed by 35 cycles of 30 s steps at 94 °C, 50 °C, and 72 °C for 90 s, and a final extension step of 72 °C for 5 min. Electrophoresis was performed (110 v–45 min) to analyze the PCR product with 1% agarose gel after amplification, and Easy ladder 1 and loading buffer (Bioline, USA) were used to see the target band. PCR products were sent to Macrogen, South Korea to execute Sanger sequencing and sequence data were analyzed using Geneious R10 software to confirm the identification of *L. plantarum* YML015, and, finally, submitted to NCBI GenBank. Homology of the selected strain *L. plantarum* YML015 was compared with other similar strains. A phylogenic tree was constructed using the Geneious version 2021.2 (Biomatters, NZ) software. In brief, a multiple sequence alignment (MSA) was conducted with the MUSCLE algorithm with default settings. The resulting MSA was used to construct a cladogram tree, using Tamura-Nei as the genetic distance model and Neighbor-Joining as the tree-building method. Resampling was conducted using bootstrap methods for random seeding, with 100 replicates, which created a consensus tree with a support threshold of 50%. Various color codes were used to visually present differences. A "similarity matrix" was also added.

### 2.2. Characterization of *L. plantarum* YML015

### 2.2.1. Biochemical Characterization

The strain was characterized biochemically using the API 50 CHL strip and API CHL medium systems, according to the manufacturer's instructions (API bioMerieux, St. Louis, MO, USA). Briefly, a freshly grown colony of *L. plantarum* YML015 was harvested and re-suspended in sterile water to achieve a cell density of $10^8$ CFU/mL. A 2 mL aliquot of the cell suspension was inoculated into 10 mL of APL 50 CHL medium and mixed gently by inversion. Further, 120 μL of this suspension was inoculated into API 50 CHL strips that were overlaid with mineral oil and were incubated for 48 h before reading the color changes [31].

### 2.2.2. Heat Stability Test

In order to confirm stability and antiviral activity of *L. plantarum* YML015 against IFVA H1N1, 1-fold and 10-fold CFS of *L. plantarum* YML015 were subjected to different heat treatments through exposure to varying degrees of temperature, such as 30 °C, 45 °C, 60 °C, 90 °C,

and 121 °C with different time durations, such as 5 min, 10 min, and 15 min. The antiviral activity of heat-treated 10-fold CFS was determined by cytopathogenic reduction assay.

### 2.2.3. Antibiotic Susceptibility Assay

The antibiotic susceptibility of *L. plantarum* YML015 was carried out using the micro-dilution method [32]. Prior to the assay, *L. plantarum* YML015 was pre-cultured at 37 °C for 24 h and subjected to LAB susceptibility test medium (LSM) formulation consisting of a mixture of IST broth (90%) and MRS broth (10%) and adjusted to pH 6.7. After 24 h of incubation, the inoculum was prepared, and a colony was suspended in a sterile plastic culture tube containing 2 mL of sterile saline until a density corresponding to a McFarland (McF) standard of 1 or a spectrophotometric equivalent ($3 \times 10^8$ CFU/mL) was obtained. The inoculated saline suspension was diluted at 1:500 to obtain a final concentration of $3 \times 10^5$ using LSM broth [33]. According to the European Food Safety Authority (EFSA) guidelines, the following antibiotics were tested at the concentration range (mg/L) given in parenthesis: gentamicin (0.5 to 256 mg/L; Tokyo chemical industry co; Ltd., Tokyo, Japan), kanamycin (2 to 1024 mg/L; Biopure reagent, Seoul Korea), streptomycin (0.5 to 256 mg/L; Tokyo chemical industry co; Ltd., Tokyo, Japan), tetracycline (0.125 to 64 mg/L; Tokyo Chemical Industry Co; Ltd., Tokyo, Japan), erythromycin (0.016 to 8 mg/L; Tokyo Chemical Industry co; Ltd., Tokyo, Japan), clindamycin (0.032 to 16 mg/L; Sigma-Aldrich, St. Louis, MO, USA), chloramphenicol (0.125 to 64 mg/L; Sigma-Aldrich, St. Louis, MO, USA), and ampicillin (0.032 to 16 mg/L; Biopure reagent, Seoul, Korea). Further, 50 μL of $3 \times 10^5$ CFU/mL inoculum was added to each well (already containing antibiotics) in a micro-dilution plate within 30 min of the preparation of the standardized inoculum. The well without antibiotics served as positive controls but contained mixture of the test strain and the medium containing solvent, which was used to dissolve the antibiotics at the highest concentration, while wells without the test strain and the antibiotics but with the medium, served as negative controls. The susceptibility panel was incubated anaerobically at 37 °C for between 24 and 48 h after inoculation; growth in the susceptibility panel was evaluated visually by comparing the pellet at the bottom of each well, comparing with the positive and negative controls. The selection of antibiotics and the characterization of sensitivity/resistance were conducted following the guidelines and breakpoints of the European Commission [34] and the European Food Safety Authority [35].

### 2.2.4. Hemolytic Phenomenon Assay

The hemolytic phenomenon assay was performed according to Maragkoudakis et al., 2009 [36]. *L. plantarum* YML015 was freshly cultured in MRS broth and then streaked on tryptone soya agar (Oxoid, Basingstoke, UK) containing 5% (*v/v*) sheep blood. The plates underwent 48 h of micro aerobic incubation (37 °C, 8% $CO_2$) and were examined for hemolytic reaction. Hemolytic activity was confirmed by the presence of a clear zone around the bacterial colony. The experimental strain that produced green-hued zones around the bacterial colonies (α-hemolysis) or did not produce any effect on the blood plates (γ-hemolysis) were considered non-hemolytic. Any strains that displayed blood lysis zones around the bacteria colonies were classified as hemolytic (β-hemolysis).

### 2.3. Culture of MDCK Cell

Madin–Darby canine kidney (MDCK) cell cultures essential for supporting viral infectivity of the test virus were used in this study. MDCK cells (KTCC® AC30015) were grown with high-glucose Dulbecco's modified Eagle's medium (DMEM) (Sigma-Aldrich, St. Louis, MO, USA) supplemented with 10% heat-inactivated standard fetal bovine serum (FBS) (HyClone, Logan, UT, USA) and 1% (*v/v*) 100U/mol penicillin and 100 μg/mL streptomycin solution at room temperature (23 ± 1 °C) under normal atmospheric conditions. Cells were sub-cultured at a 1:3 ratio every 3–4 days. Briefly, media from tissue culture flasks were collected and centrifuged at $1500\times g$ rpm for 5 min. Meanwhile, a 5 mL/flask of a trypsin-versine solution (10 mL 10× trypsin (Sigma-Aldrich, St. Louis, MO, USA) pre-sterilized

versine (EDTA)) was added and incubated for 1 min to detach the cell monolayer [37]. Following cell detachment, cleaned medium was added back into the flasks to neutralize the trypsin activity. The contents of the flasks were then removed and centrifuged at $1500 \times g$ rpm for 5 min. Following centrifugation, the supernatant was removed, and new growth medium was used to re-suspend the cells. Cells, split 1:3, were placed back into flasks and total media volume was brought up to a 20–22 mL/flask. Flasks were then placed back into their respective incubators and monitored daily.

### 2.4. MTT Cell Viability Assay

The mitochondrial-dependent reduction of MTT (3-(4,5-Dimethylthiazol-2-Yl)-2,5-Diphenyltetrazolium Bromide) to formazan was used to measure cell respiration as an indicator of cell viability. Briefly, MDCK cells were seeded in 96-well plates at a density of $5 \times 10^4$ cells/well. After 24 h of incubation, the adherent cells were treated with various concentrations (CFS $1\times$, CFS $10\times$, Heat-Killed (HK) CFS $10\times$, and CFS $20\times$) of *L. plantarum* YML015. Twenty-four hours later, after changing the medium, MTT was added to a final concentration of 0.5 mg/mL, and the cells were incubated for 4 h at 37 °C and 5% $CO_2$. The medium was then removed, and the formazan precipitate was solubilized in DMSO. The absorbance was measured at 570 nm on a microplate reader (Biotek, Winooski, VT, USA) [38].

### 2.5. Virus Culture

Highly pathogenic Influenza virus (IFV) H1N1 (A/Korea/01/2009) was obtained from the Korea Disease Control and Prevention Agency, Korea. The IFVA H1N1 was incubated in the allantoic cavity of 11-day-old chicken embryos for 3 days. In brief, specific pathogen-free (SPF) chicken eggs were cleaned by 70% (*v/v*) ethanol and incubated for embrocation in a rotating and static egg incubator at 35 °C with a humidified environment for 11 days. After 11 days, a 100 μL IFVA H1N1 was inoculated in the allantoic cavity by using 1 mL syringes with 27-gauge, 1-inch or 1.5-inch hypodermic or blunt-end needles and the punched holes were sealed with wax followed by incubation at 35 °C in a humidified, static incubator for 2–3 days. At the completion of the incubation period, eggs were prepared for fluid collection by placing them on ice (at 4 °C) overnight. Then, a 3 mL syringe with a 21-gauge, 1-inch hypodermic needle (for collection of allantoic fluids) was used to collect the harvested allantoic fluid (Figure 1). Allantoic fluid was stored at −80 °C as a stock solution of the IFVA H1N1 A virus [39,40]. Virus titers in the allantoic fluid stock solution were determined as embryo infection dose $10^{5.5}EID_{50}/0.1$ mL by using the 50% end point dilution assay described by Reed and Muench, 1938 [41].

### 2.6. Antiviral Potential of L. plantarum YML015

2.6.1. Cytopathogenic Reduction Assay

For assessing the cytopathogenic reduction effect of *L. plantarum* YML015 in vitro, firstly, the MDCK cell line was cultured in a 96-well microplate using Eagle's minimal medium (MEM) with 10% (*v/v*) fetal bovine serum (FBS), 1% (*v/v*) 100U/mol penicillin, and 100 μg/mL streptomycin solution for 24–36 h at 37 °C in a 5% $CO_2$ cell culture incubator. MDCK cells were seeded onto a 96-well culture plate at a concentration of $2 \times 10^4$ cells per well calculated by a hemocytometer [42]. Sterilized cell-free supernatant (CFS) of *L. plantarum* YML015 prepared by filtration using 0.22 μm cellulose acetate filter was serially diluted two-fold with 2% FBS DMEM solution. IFVA $H_1N_1$ was treated with a two-fold dilution of CFS of *L. plantarum* YML015 ($1\times$ and $10\times$) at 37 °C in a 5% $CO_2$ cell culture incubator for 1 h. This mixture was inoculated into MDCK cells and incubated in DMEM solution with 2% FBS at 37 °C in a humidified chamber with 5% $CO_2$ for 72 h by using a multi-pipette. The plates were observed for cytopathic effect (CPE) after 24 h, 48 h, and 72 h accordingly, and the reduction in CPE was regarded as the presence of antiviral activity [28,39,43].

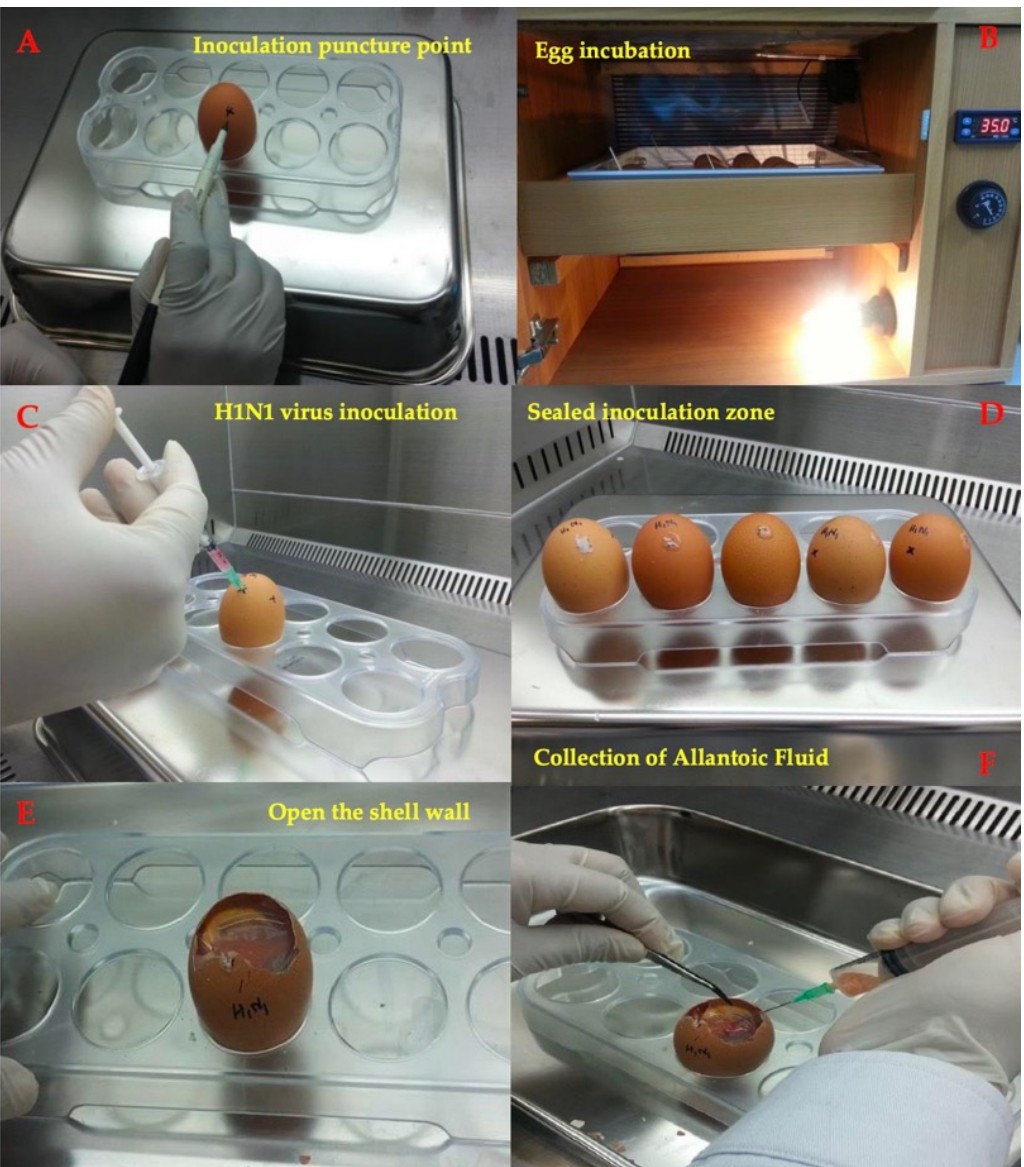

**Figure 1.** (**A**) Determine inoculation punch point; (**B**) Incubation at 35 °C, 80% humidity in egg incubator; (**C**) Inoculation of H1N1 virus for culture in allantoic fluid of 11-day-old embryo eggs; (**D**) Sealed with wax after inoculation; (**E**) After 2/3 days incubation open the egg shell wall; (**F**) Collection of allantoic fluid with freshly cultured influenza A virus HN1.

### 2.6.2. Hemagglutination Inhibition Assay

The hemagglutination inhibition assay is a widely used, inexpensive, rapid, and standard screening method for detecting activity of the target sample against IFVA H1N1. The CFS (1-fold and 10-fold) and heat-killed CFS (121 °C for 15 min, 10-fold) of *L. plantarum* YML015 were used for the hemagglutination reduction assay. The mixture of the *L. plantarum* YML015 supernatant and embryo infection dose $10^{5.5}$EID$_{50}$/0.1 mL were incubated for 1 h at 37 °C in a 5% CO$_2$ cell culture incubator and transferred into new tubes for hemagglutination inhibition assay. By using a multi-channel pipette, two-fold dilution of treated samples was made in 50 μL with PBS in a V-form-shaped 96-well microplate. Serially diluted samples were reacted with an equal volume of 1% Specific Pathogen-Free (SPF) chicken red blood cell (RBC) at room temperature for 2–3 h to allow hemagglutination inhibition reaction. The only virus dose was used to check the negative control to compare the activity of the treated sample [28,39,43].

### 2.6.3. In Ovo Antiviral Assay

Specific pathogen-free embryonated chicken eggs were used to check the in ovo antiviral activity of *L. plantarum* YML015. The experiment was designed with 6 groups, consisting of group 1: H1N1 + CFS (1-fold and 10-fold), group 2: H1N1 + HKCFS (10-fold), group 3: H1N1 + cell mass (CM), group 4: drug control H1N1 + Tamiflu, and group 5: negative control H1N1 + PBS and group. Each group contained 2 SPF eggs. After dose inoculation, eggs were incubated again for 72–96 h at 37 °C under 80% humidified incubation conditions. A similar dose of IFVA H1N1 $10^{5.5}EID_{50}/0.1$ mL was used as mentioned in previous experiments. The percentage of surviving eggs out of the total was calculated to determine the antiviral activity [44].

### 2.7. Statistical Analysis

All data were expressed as mean $\pm$ standard error of mean (S.E.M.). Statistical significance was analyzed by one-way analysis of variance (ANOVA) followed by Dunnett's post hoc test of significance. All statistical analyses were performed with Prism 4.0 (GraphPad software Inc., San Diego, CA, USA). $p < 0.05$ was considered to be significant.

## 3. Results

### 3.1. Isolation and Characterization of L. plantarum YMLO15

Among 1200 bacterial strains screened, cell-free supernatant (CFS) of the isolated bacteria *Lactobacillus* spp., isolated from Kimchi, was found to exert the highest antiviral activity, even at two-fold dilution and selected as a potential probiotic *Lactobacilli* against highly pathogenic influenza virus A (IFVA) $H_1N_1$ for further in vitro experiments. The molecular systematic analysis based on 16S rRNA sequences (Sanger sequencing) isolated from the bacterial strain *Lactobacillus* spp. displayed 100% similarity with *Lactobacillus plantarum* spp. Based on these parameters, *Lactobacillus plantarum* spp. was characterized and named *Lactoplantibacillus plantarum* YML015 (previously named *Lactobacillus plantarum* YML015) and the gene sequence was submitted to the GenBank with a nucleotide accession number KT339389. Figure 2 represents the neighbor-joining phylogenetic tree of *L. plantarum* YML015, using Geneious v10.2.3 (Biomatters, NZ). Sequences were aligned with CLUSTALW and overhanging ends were trimmed off manually.

### 3.2. Biochemical Characterization of L. plantarum YML015

The analysis of the API 50 CHL Kit for biochemical characterization showed that the strain *L. plantarum* YML015 was most closely related to *Lactoplantibacillus plantarum*, as shown in Table 1. According to API web software, the strain *L. plantarum* YML015 indicated typical carbohydrate utilization that used D-xylose, D-galactose, D-glucose, D-fructose, Dmannose, L-rhamnose, D-mannitol, D-sorbitol, N-acetylglucosamine, amygdalin, arbutin, esculin, salicin, Dcellobiose, D-maltose, D-lactose, D-melibiose, D-saccharose, D-trehalose, D-melezitose, D-raffinose, gentiobiose, D-turanose, D-tagatose, and D-arabitol. The strip capsule changed color from violet to yellow, indicating complete fermentation.

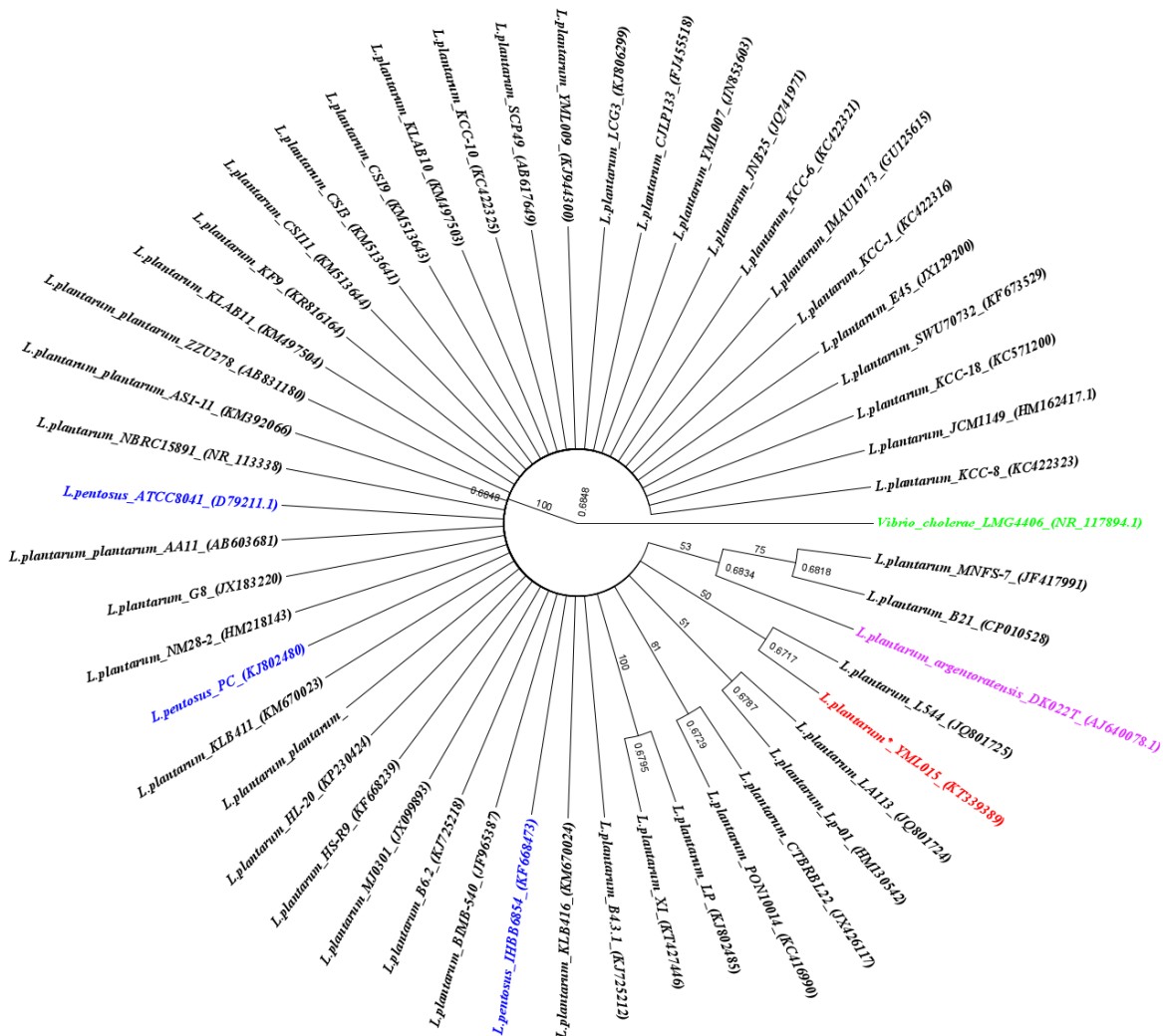

**Figure 2.** Neighbor-joining phylogenetic tree showing the position of *L. plantarum* YML015 among the different *Lactobacillus* strains based on 16s rDNA sequence.

### 3.3. Antibiotic Susceptibility Assay

The antibiotic susceptibility of *L. plantarum* YML015 to eight antibiotics is presented in Table 2. *L. plantarum* YML015 showed sensitivity to all the recommended antibiotics checked. The minimum inhibitory concentration (MIC) values were observed after 24 and 48 h of incubation, which were obtained as follows with green color in the table: gentamicin (0.5 mg/L), kanamycin (4 mg/L), streptomycin (2 mg/L), tetracycline (8 mg/L), erythromycin (0.125 mg/L) clindamycin (0.125 mg/L), chloramphenicol (4 mg/L), and ampicillin (0.25 mg/L). Since *L. plantarum* YML015 did not show any resistance to the panel of antibiotics, it could be considered as a probiotic bacterium and safe for human consumption.

### 3.4. Heat Stability Test

The temperature/heat stability of the CFS (10-fold) of *L. plantarum* YML015 was checked by using the cytopathic reduction assay. The sample treated with 121 °C for 15 min also strongly inhibited the cytopathic effect of IFVA H1N1 on the MDCK cell monolayer in a 96-well microtiter plate. The experiment was continued up to 96 h, and every 24 h later, the activity was checked (Table 3).

**Table 1.** Biochemical test by carbohydrate interpretation assay (API 50 kit).

| Biochemical Characterization of YML015 by Carbohydrate Interpretation Assay | | | |
|---|---|---|---|
| **Carbohydrates List** | **Results** | **Carbohydrates List** | **Results** |
| Control | − | Esculin | + |
| Glycerol | − | Salicin | + |
| Erythritol | − | d-cellobiose | + |
| d-arabinose | − | d-maltose | + |
| l-arabinose | + | d-lactose (bovine origin) | + |
| d-ribose | + | d-melibiose | + |
| d-xylose | − | d-saccharose | + |
| l-xylose | − | d-trehalose | − |
| d-adonitol | − | Lnulin | + |
| Methyl–β–d- xylopyranoside | + | d-melezitose | − |
| d-galactose | + | d-raffinose | − |
| d-glucose | + | Amidon (starch) | − |
| d-fructose | + | Glycogen | − |
| d-mannose | − | Xylitol | + |
| l-sorbose | − | Gentiobiose | − |
| l-rhamnose | − | d-turanose | − |
| Dulcitol | − | d-lyxose | + |
| Inositol | + | d-tagatose | - |
| d-mannitol | + | d-fuccose | − |
| d-sorbitol | − | l-fuccose | − |
| Methyl-a-d-mannopytanoside | + | d-arabitol | − |
| Methyl-a-d-glucopyranoside | + | l-arabitol | + |
| N-acetylglucosamine | + | Potassium Gluconate | − |
| Amygdalin | + | Potassium 2-Ketogluconate | − |
| Arbutin | − | Potassium 5-Ketogluconate | − |

Note: positive (+) samples were denoted by a change in color to yellow; no change in color indicated negative (−) samples.

**Table 2.** Antibiotic susceptibility assay (MIC test) of *L. plantarum* YML015.

| | | | | | =MIC | | <MIC | | >MIC | | mg/L | |
|---|---|---|---|---|---|---|---|---|---|---|---|---|
| **Antibiotics** | [A] **1** | **2** | **3** | **4** | **5** | **6** | **7** | **8** | **9** | **10** | **11** | [B] **12** |
| Ampicillin | P | 0.032 | 0.063 | 0.125 | 0.25 | 0.5 | 1 | 2 | 4 | 8 | 16 | N |
| Gentamycin | P | 0.5 | 1 | 2 | 4 | 8 | 16 | 32 | 64 | 128 | 256 | N |
| Kanamycin | P | 2 | 4 | 8 | 16 | 32 | 64 | 128 | 256 | 512 | 1024 | N |
| Streptomycin | P | 0.5 | 1 | 2 | 4 | 8 | 16 | 32 | 64 | 128 | 256 | N |
| Erythromycin | P | 0.016 | 0.032 | 0.063 | 0.125 | 0.25 | 0.5 | 1 | 2 | 4 | 8 | N |
| Clindamycin | P | 0.032 | 0.063 | 0.125 | 0.25 | 0.5 | 1 | 2 | 4 | 8 | 16 | N |
| Tetracycline | P | 0.125 | 0.25 | 0.5 | 1 | 2 | 4 | 8 | 16 | 32 | 64 | N |
| Chloramphenicol | P | 0.125 | 0.25 | 0.5 | 1 | 2 | 4 | 8 | 16 | 32 | 64 | N |

Note: Blue marked values are <MIC (mg/L); Green marked values are =MIC (mg/L); Yellow marked values are >MIC (mg/L). [A] Positive control well without antibiotics, but with the test strain containing solvent, which is used to dissolve the antibiotics at the highest concentration. [B] Negative control well without the test strain and the antibiotic, but with the medium.

**Table 3.** Temperature stability test of the cell-free supernatant (CFS) of *L. plantarum* YML015.

| Temperature Stability Test CFS (10-Fold) | | | | | |
|---|---|---|---|---|---|
| **Temperature** | **Time** | **24 h** | **48 h** | **72 h** | **96 h** |
| | 5 min | + | + | + | + |
| 30 °C | 10 min | + | + | + | + |
| | 15 min | + | + | + | + |
| | 5 min | + | + | + | + |
| 45 °C | 10 min | + | + | + | + |
| | 15 min | + | + | + | + |
| | 5 min | + | + | + | + |
| 60 °C | 10 min | + | + | + | + |
| | 15 min | + | + | + | + |
| | 5 min | + | + | + | + |
| 90 °C | 10 min | + | + | + | + |
| | 15 min | + | + | + | + |
| | 5 min | + | + | + | + |
| 121 °C | 10 min | + | + | + | + |
| | 15 min | + | + | + | + |

### 3.5. Hemolytic Phenomenon Assay

*Lactobacillus plantarum* YML015 was streaked on tryptone soy agar supplemented with 5% sheep blood and incubated in a $CO_2$ incubator at 37 °C for 48 h (Figure 3). It was found that no β-hemolysis (clear zones around colonies) was observed on the plate agar. *L. plantarum* YML015 did not produce any effect or lysis on the blood plates (γ-hemolysis) (Figure 3C). *Streptococcus aureus* KCTC 1621 was used as a negative control for β-hemolysis (Figure 3B), whereas *Lactobacillus sakei* probio65 KCTC 10755BP was used as a positive control for γ-hemolysis (Figure 3A).

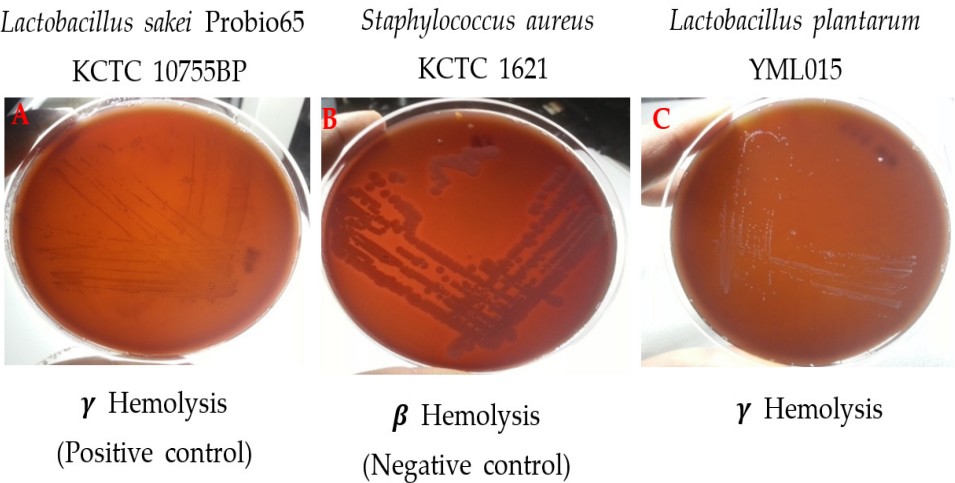

**Figure 3.** (**A**) Hemolytic phenomenon assay. Positive control: *Lactobacillus sakei* Probio 65 γ hemolysis; (**B**) Negative control *Staphylococcus aureus* KCTC 1621 β hemolysis (negative control); (**C**) Tested strain *L. plantarum* YML015 γ hemolysis.

### 3.6. MTT Cell Viability Assay

The result of MTT cell viability of the CFS of *L. plantarum* YML015 significantly showed no evidence of cytotoxicity with cell viability ranging from 80% to 100% at different concentrations (CFS 1×, CFS 10×, HKCFS 10×, and CFS 20×) compared with the standard (Figure 4). However, considering the concentration of 20× (80% cell viability rate), the current experiment shows that the cell viability of *L. plantarum* YML015 in the concentration

of $1\times$ and $10\times$ (both CFS and HKCFS) is 100% without occurring any type of cell damage or depreciation of the cell. Based on the research outcome, *L. plantarum* YML015 concentrations used for the MTT cell viability assay (cell viability >85%) were selected for the subsequent following antiviral experiments.

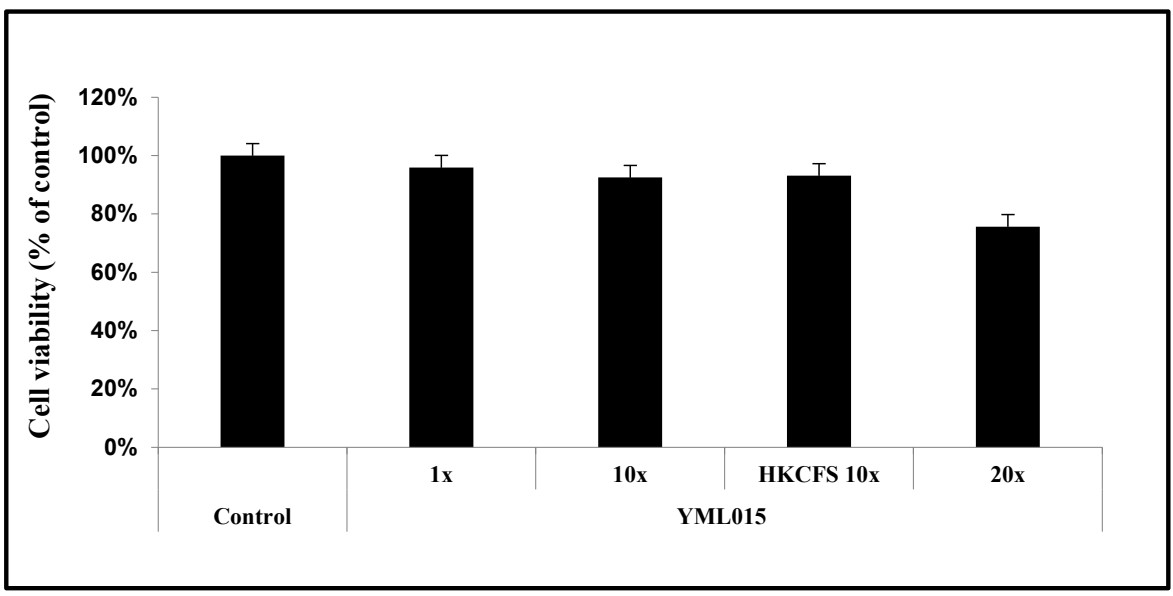

**Figure 4.** Effect of *L. plantarum* YML015 in different concentrations (CFS $1\times$, CFS $10\times$, HKCFS $10\times$, and CFS $20\times$) on MTT cell viability assay in MDCK cells with LPS induced. Cell viability was measured using the MTT method. Each determination was made in triplicate and less 5% of standard deviation error.

### 3.7. Cytopathic Reduction Assay

In vitro antiviral activity of *L. plantarum* YML015 was evaluated using MDCK cells by the cytopathic reduction assay. In this assay, *L. plantarum* YML015 was grown for its antiviral activity against influenza virus A H1N1 on MDCK cells cultured in a 96-well plate. We found that H1N1 inoculation caused a cytopathic effect in MDCK cells when used as negative control (Figure 5). However, the same cytopathic effect was reduced or not observed in MDCK cells treated with cell-free supernatant of *L. plantarum* YML015 (1-fold and 10-fold *v/v*) even after 72 h of the viral dose $10^{5.5}$EID$_{50}$/0.1 mL (Figure 5B). In addition, microscopic observation results proved that MDCK cells treated with H1N1 and *L. plantarum* YML015 showed similar morphology as control MDCK cells (without any treatment) as shown in Figure 5C. Therefore, it proved that *L. plantarum* YML015 could be considered as a potential anti-influenza bacterium able to reduce the cytopathic effect in MDCK cells. In addition, 10-fold (*v/v*) cell-free supernatant of *L. plantarum* YML015 showed more specific results. Serial dilutions of $2^0$, $2^1$, and $2^2$ of 10-fold cell-free supernatant also exhibited similar activity to control the CPE in MDCK cells (Table 4).

### 3.8. Hemagglutination Inhibition Assay

Hemagglutination effect is a common phenomenon of influenza A virus H1N1 on chicken red blood cells. This assay revealed the hemagglutination inhibition potential of the cell-free supernatant of *L. plantarum* YML 015, which can inhibit the viral HA protein that causes hemagglutination. As described in Table 5, 1-fold, 10-fold, and heat-killed (10-fold) CFS of *L. plantarum* YML015 showed anti-hemagglutination effect against H1N1. Interestingly, CFS (10-fold) and heat-killed CFS (10-fold) of *L. plantarum* YML015 showed similar hemagglutination inhibition activity of 2-fold dilution (4 times).

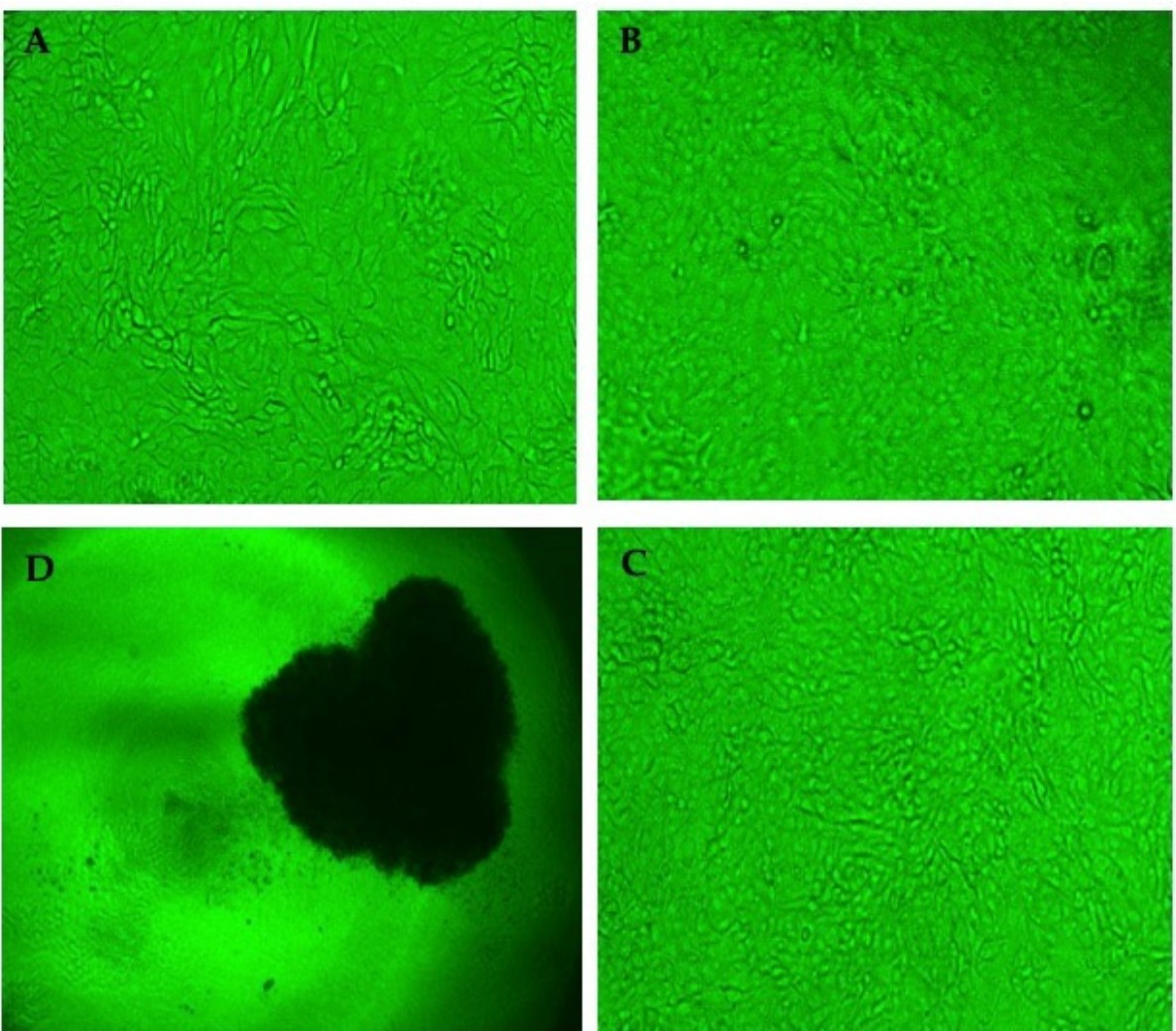

**Figure 5.** Cytopathogenic effects of IFVA H1N1 virus infection in MDCK cells. (**A**) Control cells without any treatment; (**B**) MDCK cells treated with *L. plantarum* YML015 CFS (1-fold) and H1N1l; (**C**) MDCK cells treated with *L. plantarum* YML015 CFS (10-fold) and H1N1; (**D**) cytopathogenic effects in MDCK cells treated with H1N1 (negative control). All pictures were taken by Olympus IX71 Fluorescence Electron Microscope at 10 × 0.30 Megapixel.

**Table 4.** Cytopathogenic reduction effect of CFS of *L. plantarum* YML015 against H1N1.

| Sample Name | Cytopathogenic Reduction Effect of CSF of *L. Plantarum* YML015 | | | | |
|---|---|---|---|---|---|
| | **Two-Fold Dilution** | | | | |
| | $2^0$ | $2^1$ | $2^2$ | $2^3$ | $2^4$ |
| CSF (1-fold) | + | 0 | 0 | 0 | 0 |
| CSF (10-fold) | ++ | + | 0 | 0 | 0 |

Virus titer: $10^{5.5}$ EID50/0.1 mL; (+) antiviral activity; (++) High antiviral activity; CFS: Cell-free supernatant.

**Table 5.** Hemagglutination inhibition effect of CFS of *L. plantarum* YML015 against H1N1.

| Sample Name | Hemagglutination Inhibition Effect of CSF of L. *plantarum* YML015 | | | | | | |
|---|---|---|---|---|---|---|---|
| | Two-Fold Dilution | | | | | | |
| | $2^0$ | $2^1$ | $2^2$ | $2^3$ | $2^4$ | $2^5$ | $2^6$ |
| [a] CSF (10-fold) | ++ | ++ | ++ | ++ | + | 0 | 0 |
| [b] HKCSF (10-fold) | ++ | ++ | ++ | ++ | + | 0 | 0 |

Virus titer: $10^{5.5}$ EID50/0.1 mL; (++) High hemagglutination inhibition activity; (+) Hemagglutination inhibition activity; (**0**) Hemagglutination effect of H1N1; [a] CFS: Cell-free supernatant, [b] HKCFS: Heat-killed cell-free supernatant.

### *3.9. In Ovo Antiviral Activity*

In this assay, influenza A virus H1N1 was treated with 5 groups of treatments prepared form *L. plantarum* YML015 with 1 group of negative control. As described in Table 6, for this experiment, each group contained 2 embryos obtained from SPF eggs that had been incubated for 11 days. Among 6 groups, the untreated virus control indicated a marked antiviral effect on the test virus. Heat-killed CFS (10-fold) and only CFS (10-fold) treated eggs showed a 100% survival rate. Apart from this, the survival rate was 50% in the case of the positive control treated by Tamiflu (Oseltamivir phosphate). Treatment with cell mass $(3 \times 10^8$ cfu/mL) and CFS (1-fold) also gave similar results as with the positive control.

**Table 6.** In ovo antiviral activity of *L. plantarum* YML015 on embryo eggs.

| Antiviral Activity of *Lactobacillus plantarum* YML015 Using Specific Pathogen-Free (SPF) Eggs | | | |
|---|---|---|---|
| Group | Dilution | Survival/Total | % |
| IFVA (H1N1) + CFS [a] | $1{:}10^4$ | 1/2 | 50% |
| IFVA (H1N1) + CFS (10 fold) [b] | $1{:}10^4$ | 2/2 | 100% |
| IFVA (H1N1) + HKCFS (10 fold) [c] | $1{:}10^4$ | 2/2 | 100% |
| IFVA (H1N1) + CM [d] | $1{:}10^4$ $(3 \times 10^8$ CFU/mL) | 1/2 | 50% |
| IFVA (H1N1) + T [e] | 5 mg/0.1 mL | 1/2 | 50% |
| IFVA (H1N1) + PBS [f] | $1{:}10^4$ | 0/2 | 0% |

Influenza virus A (IFVA) titer: $10^{5.5}$ EID50/0.1 mL; [a] CFS: Cell-free supernatant; [b] CFS (10-fold): Cell free supernatant (10-fold); [c] HKCFS (10 fold): heat-killed cell-free supernatant (10-fold); [d] CM: Cell Mass; [e] T: Tamiflu (Tamiflu® capsule, containing oseltamivir phosphate 98.5 mg equivalent to oseltamivir 75 mg); [f] PBS; phosphate buffer solution pH 7.4.

## 4. Discussion

Nowadays, the use of probiotics is in high demand due to their high immunostimulant potency. For example, the probiotic strain *Lactobacillus plantarum* KLAB21 from Korean kimchi has significant antimutagenic characteristics [45,46]. There is a huge trend among researchers to explore and study different probiotics or Lactic Acid Bacteria (LAB) from Korean fermented food products. Studies have also observed their chemical nature, along with antimicrobial potential against various food spoilage and food-born pathogens, including molds and fungi, for their possible use as food preservatives [29]. In this current study, a total of 1200 LAB strains were isolated from different kinds of Kimchi samples, which were initially screened for their antiviral efficacy. Among them, one of the strains designated as *L. plantarum* YML015 exhibited remarkable antiviral activity against IFVA H1N1. The 16 s rRNA sequencing confirmed molecular characterization of the YML015 isolate as *L. plantarum* YML015. The selected strain was also further identified based on the biochemical characterization utilizing carbohydrate interpretation assay (API 50CHL kit). A standard method of carbohydrate utilization according to API web software showed that *L. plantarum* YML015 used approximately 24 carbohydrates among 50 carbohydrates (Table 1), thus, was characterized as Gram-positive and *lactobacilli*.

In our study, the probiotic nature of *L. plantarum* YML015 was confirmed based on the susceptibility test result. However, the European Food Safety Association (EFSA) has

already published a new protocol for foods and feeds purposes, which might be helpful to evaluate probiotic strains as a food grade probiotic [47]. Moreover, "Qualified Presumption of Safety" (QPS) also described the importance and procedure of determination of antibiotic resistance for selected probiotic strains. We experienced that the probiotic strain *L. plantarum* YML015 was very sensitive to clinically essential antibiotics such as ampicillin, gentamycin, kanamycin, streptomycin, erythromycin, clindamycin, and tetracycline. However, phenotypic antibiotic resistance testing in LAB requires standardization as it can be subjected to species specificity but is also dependent on the method and the media used [48]. Eventually, *L. plantarum* YML015 was found to cause γ-hemolysis (non-hemolysis). Results proved again that *L. plantarum* YML015 might be a potential probiotic with non-hemolytic characteristics. It is also a common phenomenon of non-hemolytic LAB isolates from food sources [49,50], except some strains of *Enterococcus faecalis*. Furthermore, Table 2 describes temperature stability of the cell-free supernatant of *L. plantarum* YML015. Interestingly, at optimum temperature treatment of 121 °C for 15 min, antiviral activity of CFU of *L. plantarum* YML015 was the same up to 96 h of activity observation. In addition to the MTT cell viability study, it has been significantly demonstrated that *L. plantarum* YML015 has optimum cell viability percentage with no cell damage or cytotoxic effect (Figure 4). The result supported that, as a probiotic, *Lactobacillus plantarum* YML015 is highly safe for human and animal consumption. The study also found that heat-killed bacteria, especially heat-treated probiotic strains, showed immunomodulating effects and had antagonizing properties against pathogens [51]. Hence, it was confirmed that *L. plantarum* YML015, as a strong probiotic, could be used as a heat-stable and effective agent against the H1N1 virus.

Viral invasion causes structural damage or change in the host cells called cytopathic effect (CPE). Two kinds of effects might occur due to CPE, such as the infecting virus leading to lysis of the host cell or because of the inability to reproduce new cells (cell death). It is also called cytopathogenic when morphological changes in the host cell are caused by virus. Moreover, [42,52] described that probiotic *Lactobacilli* strains exhibited antiviral effect against influenza virus as experienced with the cytopathic reduction assay. In our experiment, we successfully confirmed that newly invented *L. plantarum* YML015 has significant potential to inhibit the cytopathic effect caused by IFVA H1N1. It was ensured that the cell-free supernatant (1-fold and 10-fold) of *L. plantarum* YML015 completely suppressed the cytopathic effect of IFVA H1N1 on a monolayer of the MDCK cell line as compared with the negative control as described in Figure 5.

The hemagglutination inhibition assay is a popular screening assay. It allows rapid detection of antiviral activity of any kinds of sample against influenza virus [33,53]. This assay was standardized on the basis of the ability of viral HA protein to bind and to coagulate/agglutinate human or chicken red blood cells (RBC) [39]. Our findings are aligned with the previous observations, and it was ascertained in our research work that CFS of *L. plantarum* YML015 exhibited most effective potential in the hemagglutination inhibition activity assay (Table 5). This strongly suggests that probiotic dietary uptake of *L. plantarum* YML015 could be beneficial for direct inhibition of influenza virus A H1N1 infection [42].

In addition, this research also assessed the in ovo potential of the probiotic strain *L. plantarum* YML015 to reconfirm the antiviral efficacy on influenza A virus H1N1 and was expressed as a percentage of survival from the total number of test samples. Among all treated samples, including the negative control, only heat-killed CFS (10-fold) and CFS (10-fold) showed highest antiviral activity compared to other tested samples, and the survival rate was 100%. However, inoculation of eggs with standard treatment (Tamiflu) showed a 50% survival rate, suggesting that as a probiotic, *L. plantarum* YML015 could be a good source of natural anti-influenza drugs.

## 5. Conclusions

In summary, the present research output highlights the significant therapeutic efficiency of the probiotic strain *L. plantarum* YML015 isolated from the Korean fermented



food Kimchi as a successfully competent antiviral agent against extremely pathogenic influenza A virus H1N1. Throughout the in vitro and in ovo experiments, it was clearly evident that the cell-free supernatant (CFS) of *L. plantarum* YML015 showed remarkable performance to kill or inhibit the activity of influenza A virus H1N1 on the MDCK cell line. The present research findings have further increased the urgency for clinical studies and animal trials to ensure that *L. plantarum* YML015 could be a safe natural probiotic drug with high therapeutic potential against deadly influenza virus H1N1 as compared with available bioorganic or synthetic drugs.

**Author Contributions:** Conceptualization, R.M. and Y.-H.P.; methodology, R.M. and M.B.A.; software, R.M.; formal analysis, R.M.; investigation, R.M.; resources, Y.-H.P.; writing—original draft preparation, R.M. and M.B.A.; writing—review and editing, S.-H.L., K.R.P., H.P.D., K.A.A. and P.M.H.; supervision, Y.-H.P.; project administration, Y.-H.P.; funding acquisition, Y.-H.P. All authors have read and agreed to the published version of the manuscript.

**Funding:** This research received no external funding.

**Institutional Review Board Statement:** Not applicable.

**Informed Consent Statement:** Not applicable.

**Data Availability Statement:** The data presented in this study are available upon request from the corresponding author.

**Acknowledgments:** This research was supported by the Yeungnam University Research Assistantship (RA) given to Rajib Majumder.

**Conflicts of Interest:** The authors declare no conflict of interest.

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
