# Peer review of "Anti-Influenza Virus Potential of Probiotic Strain Lactoplantibacillus plantarum YML015 Isolated from Korean Fermented Vegetable"

_fermentation, doi:10.3390/fermentation8110572_

Round 1

Reviewer 1 Report

This paper shows good work all sections are well described except the discussion part is very poor, all figures and tables should be more clarified.

The authors should show at least the basis of selection of this strain based on table or figure data for the best obtained isolates amongst the 1200 obtained,  (best screening data on numerical values even as the supplementary files)

L30

;

Change to observed    Ëƒ to observe

L54

;

Change but also ˃ beside

L57

;

Correct consist to consisted

L60

;

Correct lactobacillus

L70-72

;

Rephrase  to be clear

L72

;

Correct influenza

L96, L111

;

Correct Lactobacillus plantarum format

L127

;

Correct product

L145

;

Replace was  by were

L154, l235, L243

;

Correct oc

L180

;

Correct to [36]  and add the authors name before reference number

L195-196

;

Rephrase to be understandable

L221

;

Incubated

L231

;

Correct by 41

L288

;

Correct to(…. , the isolated bacterial…)

Figure 2

;

All scientific names should be italic font , correct this figure

L307

;

Show the color obtained for esculine

L309

;

API 50 CHL kit

Table 1

;

Correct all L and d format to be small caps, and correct the table heading  not to be active ingredient, in the footnote, what does it mean by present and absent. Please rephrase

Table 2

;

Is unclear, please reconstruct, Illustrate the numbers meaning, beside there is no green color in the table, what does it mean by yellow color and blue ? what is it s significance to the reader??? clarify

Figure 3

;

Please enhance this figure, provide high resolution pictures, I couldn’t distinguish  colors or clear zone in this fig, can be moved to supplementary after providing high resolution /clear images

L356

;

what does  it mean by plantarum yml015 concentration? Which concentration??

Fig. 5

;

Provide high resolution photos, all are unclear and correct the order, explain the colors and blacking in fig.d

             Table 4 A and B

;

Please change to 4 and 5 and reconstruct these tables to be more clear, no need for much abbreviations in the tables, provide full description in columns to clarify the tables as much as possible

Table 5

;

What is CM , not described in the footnote? ; correct 10fold;

Please show the survival ration in percentage for more clear data

Discussion

;

Don’t start sentence with numbers of references please see L422   [45,46] observed……, L476   same please rephrase; gram should be corrected to Gram

Discussion

;

The discussion part is very week, just as repetition of the result descriptions without explaining or in-depth discussion. This section should be completely rephrased to be in depth and to explain results obtained in this study

L453

;

Explain the reasons?

References

;

The authors are advised to go through all references and check for several errors  in scientific names formatting (see for example ref. 13, 36, 37…etc and errors in journal names as the first letter should be capitalized ( see for example ref. 3,6,13,14,16,17,18,19,22,23,24,25,27,28,30,36,37,41,48,50,51,….etc

Author Response

The authors are very thankful to the Editor, the Editorial team, and the Reviewers for consideration of our manuscript and for providing their valuable suggestions.

We have now addressed all the comments and incorporated the required changes into a revised version (highlighted in red font) as described in the point-by-point response below. These changes have helped to improve the manuscript substantially, which, we now hope is acceptable for publication in the MDPI Fermentation.

Reviewer 1 comments

We thank the reviewer for taking the time to comprehensively assess our manuscript and recommendations for the revisions, that has surely resulted in improvements overall.

This paper shows good work all sections are well described except the discussion part is very poor, all figures and tables should be more clarified.

Response

The authors should show at least the basis of selection of this strain based on table or figure data for the best obtained isolates amongst the 1200 obtained, (best screening data on numerical values even as the supplementary files)

We have isolated around 1200 bacterial isolates, and we did preliminary screening (anti-viral activity) of few promising bacterial strains isolated form Kimchi and we choose L. planatrum (YML015) (data not shown)

L30

;

Change to observed    Ëƒ to observe

Changed as suggested

L54

;

Change but also Ëƒ beside

Changed as suggested

L57

;

Correct consist to consisted

Changed as suggested

L60

;

Correct lactobacillus

Changed as suggested

L70-72

;

Rephrase  to be clear

We have deleted complex phrase in the sentence and made it simpler and clearer.

“Based on two specific proteins called hemagglutinin (HA) and neuraminidase (NA), there are different subtypes of influenza viruses”

L72

;

Correct influenza

influenza is changed to Influenza

L96, L111

;

Correct Lactobacillus plantarum format

Font changed to italic

L127

;

Correct product

The font P of Product has been changed to lowercase.

L145

;

Replace was by were

Was is replaced with were

L154, l235, L243

;

Correct oc

The °c (degree centigrade) were not uniform throughout the manuscript. We have now made it uniform “°C”

L180

;

Correct to [36] and add the authors name before reference number

We have not added “Maragkoudakis et al., 2009” before reference [36]

L195-196

;

Rephrase to be understandable

We have now rephrase to ”Briefly, media from tissue culture flasks were collected and centrifuged at 1500 rpm for 5 minutes” by changing “TC-75cm2” to “tissue culture”

L221

;

Incubated

The word grown is replaced with incubated

L231

;

Correct by 41

We have not corrected as “by Reed and Muench 1938”

L288

;

Correct to (…. , the isolated bacterial…)

Corrected as suggested

Figure 2

;

All scientific names should be italic font , correct this figure

Corrected as suggested

L307

;

Show the color obtained for esculine

We observed the colour changes to yellow

L309

;

API 50 CHL kit

Corrected as suggested

Table 1

;

Correct all L and d format to be small caps, and correct the table heading not to be active ingredient, in the footnote, what does it mean by present and absent. Please rephrase

Corrected as suggested

Table 2

;

Is unclear, please reconstruct, Illustrate the numbers meaning, beside there is no green color in the table, what does it mean by yellow color and blue ? what is it s significance to the reader??? clarify

Corrected as suggested

Figure 3

;

Please enhance this figure, provide high resolution pictures, I couldn’t distinguish colors or clear zone in this fig, can be moved to supplementary after providing high resolution /clear images

Corrected as suggested

L356

;

what does it mean by plantarum yml015 concentration? Which concentration??

L plantarum YMl015 concentration means the concentration of the culture medium. After centrifuge we have collected the supernatant and used 1x, 10x 7 20x fold of concentration to check the effect.

Fig. 5

;

Provide high resolution photos, all are unclear and correct the order, explain the colors and blacking in fig.d

Corrected as suggested

             Table 4 A and B

;

Please change to 4 and 5 and reconstruct these tables to be more clear, no need for much abbreviations in the tables, provide full description in columns to clarify the tables as much as possible

Corrected as suggested

Table 5

;

What is CM, not described in the footnote? ; correct 10fold;

Please show the survival ration in percentage for more clear data

CM= Cell Mass, Revised accordingly

Discussion

;

Don’t start sentence with numbers of references please see L422   [45,46] observed……, L476   same please rephrase; gram should be corrected to Gram

We have now edited those section in a way not to start sentence with number of references. For both [45,46] and L476.

gram is changed to Gram.  

Discussion

;

The discussion part is very week, just as repetition of the result descriptions without explaining or in-depth discussion. This section should be completely rephrased to be in depth and to explain results obtained in this study

This section has been revised accordingly.

L453

;

Explain the reasons?

We experienced that probiotic strain Lactobacillus plantarum YML015 was found very sensitive to clinically essential antibiotics, therefore it is very safe for human consumption as a Probiotics.

References

;

The authors are advised to go through all references and check for several errors  in scientific names formatting (see for example ref. 13, 36, 37…etc and errors in journal names as the first letter should be capitalized ( see for example ref. 3,6,13,14,16,17,18,19,22,23,24,25,27,28,30,36,37,41,48,50,51,….etc

All the suggestion has been implemented.

Reviewer 2 Report

1. I noticed that some spellings and symbols are not uniform throughout the text. These errors need to be fixed. For example H1N1 (in line 25 vs. 31), °C (in line 118 vs. 124 vs. 147), ml or μl (in line 100 vs. 223 vs. 156). In some places, "C" was written with a lowercase letter. Line 96 - Lactobacillus plantarum should be italic.

2. Why did you choose this strain from 1200 isolates? - it's not clear. Is it significantly different from others?  It's better to show data.

3. The qualities of the tables are low, did you give them as photos?

4. Most of the references are from the 2000s, the authors avoided using references from the last 5 years. The article should have been written with more recent references.

Author Response

The authors are very thankful to the Editor, the Editorial team, and the Reviewers for consideration of our manuscript and for providing their valuable suggestions.

We have now addressed all the comments and incorporated the required changes into a revised version (highlighted in red font) as described in the point-by-point response below. These changes have helped to improve the manuscript substantially, which, we now hope is acceptable for publication in the MDPI Fermentation.

Reviewer 2 comments

  1. I noticed that some spellings and symbols are not uniform throughout the text. These errors need to be fixed. For example H1N1 (in line 25 vs. 31), °C (in line 118 vs. 124 vs. 147), ml or μl (in line 100 vs. 223 vs. 156). In some places, "C" was written with a lowercase letter. Line 96 - Lactobacillus plantarumshould be italic.

Author response: We apologize for the spelling’s mistakes and inconsistency with the symbols. We have now fixed this error throughout the manuscript

  1. Why did you choose this strain from 1200 isolates? - it's not clear. Is it significantly different from others?  It's better to show data.

Author response: We did preliminary screening (anti-viral activity) of few promising bacterial strains isolated form Kimchi. Out of few most promising strains, we finally choose L. planatrum (YML015) for the further investigations. The preliminary data of 1200 isolates would be a huge dataset. So, we haven’t shown it in the manuscript.

  1. The qualities of the tables are low, did you give them as photos?

Author response: Yes, we have presented the table as photos. Based on our previous experience of publishing with MDPI, MDPI journal changes all table into screenshot (JPEG) and published it online. Therefore, we changed it to photo. However, we also agree with reviewer that its quality is low. Therefore, in the revised manuscript, we have replaced those low-quality photos with higher resolution and clear photos. During, galley proof correction, if the journal still asks us to submit the table as such (no photo), we can do it.

  1. Most of the references are from the 2000s, the authors avoided using references from the last 5 years. The article should have been written with more recent references.

Author response: We have replaced some of the very old reference with most recent and relevant one in the revised manuscript.